# Quantile $Q$-Learning: Revisiting Offline Extreme $Q$-Learning with Quantile Regression

**Xinming Gao**[*]                                                                          *seu24yoki@foxmail.com*
*School of Mathematics, Southeast University*

**Shangzhe Li**[*]                                                                          *shangzhe@unc.edu*
*Department of Computer Science, University of North Carolina at Chapel Hill*

**Yujin Cai**                                                                          *101300480@seu.edu.com*
*School of Mathematics, Southeast University*

**Wenwu Yu** [†]                                                                          *wenwuyu@seu.edu.cn*
*School of Mathematics, Southeast University*

**Reviewed on OpenReview:** *https://openreview.net/forum?id=tBKznsUimN*

## Abstract

Offline reinforcement learning (RL) enables policy learning from fixed datasets without further environment interaction, making it particularly valuable in high-risk or costly domains. Extreme $Q$-Learning (XQL) is a recent offline RL method that models Bellman errors using the Extreme Value Theorem, yielding strong empirical performance. However, XQL and its stabilized variant MXQL suffer from notable limitations: both require extensive hyperparameter tuning specific to each dataset and domain, and also exhibit instability during training. To address these issues, we proposed a principled method to estimate the temperature coefficient $\beta$ via quantile regression under mild assumptions. To further improve training stability, we introduce a value regularization technique with mild generalization, inspired by recent advances in constrained value learning. Experimental results demonstrate that the proposed algorithm achieves competitive or superior performance across a range of benchmark tasks, including D4RL and NeoRL2, while maintaining stable training dynamics and using a consistent set of hyperparameters across all datasets and domains.

## 1 Introduction

Deep reinforcement learning (DRL) has achieved impressive results across a broad range of domains, including navigation (Mirowski et al., 2018), healthcare (Yu et al., 2021a), robotics (Haarnoja et al., 2018), and games (Mnih et al., 2015; Silver et al., 2016). Recent advances in offline reinforcement learning (offline RL) (Kumar et al., 2020; Levine et al., 2020; Kostrikov et al., 2021; Garg et al., 2023) have extended the capability of DRL by enabling agents to learn solely from static datasets, without requiring further interaction with the environment. This paradigm shift is particularly promising in domains where data collection is expensive, risky, or impractical.

Among the recent developments, the offline version of extreme Q-learning (XQL) (Garg et al., 2023) introduces a novel perspective by modeling the Bellman error distribution using the Extreme Value Theorem (EVT), assuming it follows a Gumbel distribution. This theoretical insight leads to an in-sample learning algorithm that has demonstrated competitive performance on standard offline RL benchmarks such as D4RL. However, XQL exhibits notable instability during training, as reported in a follow-up work called MXQL

---

[*]These authors contributed equally.
[†]Corresponding Author: Wenwu Yu wenwuyu@seu.edu.cn

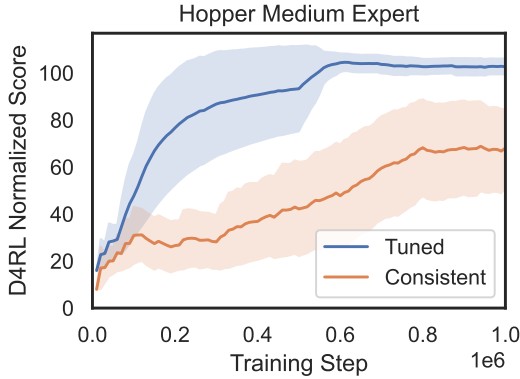 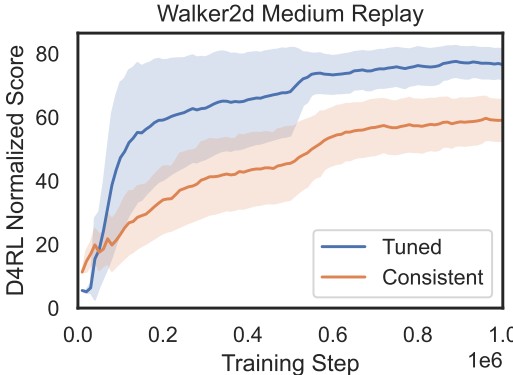

Figure 1: Comparison of XQL performance using dataset-specific tuning versus consistent hyperparameters across the domain. In some scenarios, performance degrades notably without dataset-specific tuning.

(Omura et al., 2024). To address this issue, MXQL (Omura et al., 2024) introduces a variant that stabilizes training through a Maclaurin-series-based approximation of the original XQL loss.

Despite these efforts, several challenges remain. Both XQL and MXQL are sensitive to hyperparameter choices—particularly the temperature coefficient—which often require dataset-specific tuning to achieve optimal performance, making them impractical for real-world applications. This sensitivity can lead to significant performance degradation when hyperparameters are changed, as demonstrated in Figure 1. Besides, even the stabilized MXQL variant can experience training instability under certain conditions (see Figure 2a). These limitations motivate our central research question:

> *Can we design an algorithm based on offline XQL that maintains stability and strong performance across diverse domains and offline datasets, using a single, consistent set of hyperparameters?*

To address this issue, we propose a principled method for estimating the temperature coefficient $\beta$ via quantile regression under mild assumptions, thereby eliminating the need for dataset-specific tuning. Building on this, we introduce a value regularization approach with mild generalization, inspired by prior work on doubly constrained value learning (Mao et al., 2024). The performance of the proposed algorithm matches or exceeds that of the state-of-the-art model-free offline RL algorithms across a variety of domains and dataset types in diverse offline RL benchmarks, including D4RL (Fu et al., 2020) and NeoRL2 (Gao et al., 2025), without the need for domain-specific hyperparameter tuning.

## 2 Related Works

### 2.1 Offline Reinforcement Learning

Our work builds primarily on the literature of offline reinforcement learning (offline RL), a subfield of reinforcement learning that aims to learn optimal policies from a fixed dataset without any further interaction with the environment. A central challenge in offline RL is that many online off-policy methods tend to underperform due to extrapolation error (Fujimoto et al., 2019) or distributional shift (Levine et al., 2020). To address these issues, offline RL algorithms often augment standard off-policy methods with a penalty term that measures the divergence between the learned policy and the behavior policy used to collect the data (Fujimoto & Gu, 2021). Existing offline RL approaches vary in how they formulate the problem, including methods that constrain the learned policy to remain close to the behavior policy (Zhang et al., 2023; Fujimoto & Gu, 2021; Li et al., 2022), approaches that incorporate pessimistic value estimation to avoid overestimation of out-of-distribution actions (An et al., 2021; Yu et al., 2021b; Kostrikov et al., 2021; Kumar et al., 2020), and techniques that leverage deep architectures such as large autoregressive models

(Chen et al., 2021; Janner et al., 2021) or generative models (Janner et al., 2022; Ajay et al., 2022; Wang et al., 2022; Zhang et al., 2025).

In our work, we specifically focus on improving an existing offline RL method, Extreme $Q$-Learning (XQL) (Garg et al., 2023), which is potentially unstable and sensitive to hyperparameters as discussed in the introduction. Implicit value regularization (IVR) was later proposed as a new paradigm that reinterprets XQL through an in sample value regularization perspective and derives Exponential $Q$-Learning (EQL) as an equivalent formulation (Xu et al., 2023). However, the authors also note that EQL introduces exponential terms in the value updates, which can lead to highly unstable gradients and pronounced sensitivity to hyperparameter choices in practice (Xu et al., 2023). Prior work (Omura et al., 2024) proposed a method to stabilize XQL by applying a Maclaurin expansion to the original objective function. However, this MXQL variant still relies on a manually tuned temperature parameter $\beta$ and therefore does not address the fundamental challenge of selecting $\beta$ in a robust and data driven manner.

## 2.2 Quantile Regression in Reinforcement Learning

Quantile regression serves as a foundational framework in distributional reinforcement learning. Its practical application was popularized by QR-DQN (Dabney et al., 2018b), which approximates the return distribution using a discrete set of quantiles learned through quantile-regression losses. Building upon this foundation, subsequent methods have sought more flexible and stable representations of the return quantile function. These include Implicit Quantile Networks (IQN) and non-crossing variants that explicitly enforce monotonicity across quantile levels (Dabney et al., 2018a; Zhou et al., 2020). More recently, the implicit expectile-quantile network (IEQN) (Jullien et al., 2025) extends IQN with a dual expectile-quantile regression objective that jointly learns quantiles and expectiles of the return distribution, provably converging to the true value distribution in the limit of infinite estimated quantile and expectile fractions. In this spirit, our method also employs quantile regression to parameterize an implicit value function.

## 3 Preliminaries

The problem is formulated as an Markov decision process (MDP), which is represented by the tuple $(\mathcal{S}, \mathcal{A}, \mathcal{P}, r, \gamma)$, where $\mathcal{S}$ denotes the state space, $\mathcal{A}$ represents the action space, $\mathcal{P} : \mathcal{S} \times \mathcal{A} \to \Delta_{\mathcal{S}}$ characterizes the transition dynamics; $r : \mathcal{S} \times \mathcal{A} \to \mathbb{R}$ is the reward function; $\gamma \in [0, 1)$ is the discount factor. We aim to learn a policy $\pi : \mathcal{S} \to \Delta_{\mathcal{A}}$ to maximize cumulative discounted rewards (Sutton & Barto, 2018). In offline RL setting, the policy learning is conducted on a dataset $\mathcal{D}$ consisting a set of trajectories $(\tau_1, ..., \tau_T)$. Each trajectory is composed of a group of transitions $\{(s_t, a_t, r_t, s'_t)\}$.

**Advantage Weighted Regression** One class of off-policy methods updates the policy using value functions for weighted regression, commonly referred to as Advantage Weighted Regression (AWR) (Peng et al., 2019). In AWR, the policy is updated according to:

$$\pi(a \mid s) \propto \exp\left(\frac{Q(s,a) - V(s)}{\beta}\right) \mu(a \mid s),$$

where $\mu$ denotes the behavior policy and the temperature parameter $\beta$ controls the degree of conservatism. Empirically, prior works (Xu et al., 2023; Park et al., 2024) show that AWR is highly sensitive to the choice of $\beta$: smaller values can lead to overfitting or instability, while larger values produce overly conservative policies.

**Extreme $Q$-Learning** Extreme $Q$-Learning (XQL) (Garg et al., 2023) is an algorithm capable of solving both online and offline RL tasks. It directly models the maximal value using Extreme Value Theory (EVT) with the assumption that the error in $Q$-functions follows a Gumbel distribution. In our work, we especially focus on the offline part of XQL. Specifically, offline XQL leverages an objective for optimizing a soft value function $V$ over dataset $\mathcal{D}$:

$$\mathcal{J}(V) = \mathbb{E}_{(s,a)\sim\mathcal{D}} \exp\left((Q(s,a) - V(s))/\beta\right) - (Q(s,a) - V(s)/\beta) - 1, \tag{1}$$

where $Q$ is the $Q$-function optimized via minimizing mean-squared Bellman error and $\beta$ is the temperature hyperparameter. Furthermore, XQL learns a policy with an advantage-weighted regression (Peng et al., 2019; Nair et al., 2020) (AWR) style update:

$$\pi^* = \arg\max_\pi \mathbb{E}_{(s,a)\sim\mathcal{D}} \left[ e^{(Q(s,a)-V(s))/\beta} \log \pi \right]. \tag{2}$$

In practice, we find that XQL is sensitive to the hyperparameter $\beta$, and its optimal value varies significantly across environments and offline datasets, as shown in Appendix F.1 and Figure 1. Revisiting Eqs. 1 and 2, we observe that the temperature parameter $\beta$ in the exponential terms plays a critical role in the stability of both value and policy updates, and is likely a key source of the hyperparameter sensitivity observed in XQL.

## 4 Revisiting Extreme $Q$-Learning with Quantile Regression

### 4.1 Estimating $\beta$ under Mild Assumptions

Since the formulation of XQL is sensitive to the hyperparameter $\beta$, we show in this section that $\beta$ can be generalized to a state-dependent function $\beta(s)$ and learned under mild assumptions. We begin by stating the assumptions used in our subsequent analysis involving the estimated $Q$-function $Q$, the optimal $Q$-function $Q^\star$ and the soft value function $V$. It is worth noting that these assumptions are either identical or closely aligned with those used in prior works (Hui et al., 2023; Garg et al., 2023).

**Assumption 1** ((Hui et al., 2023)). *Given a state-dependent function $\beta(s)$ and a heteroscedastic Gumbel noise $\epsilon(s,a) \sim \mathcal{G}(0, \beta(s))$:*

$$\epsilon(s,a) = Q^\star(s,a) - Q(s,a).$$

**Assumption 2.** *Given a state-dependent Gumbel noise model $\mathcal{G}(0, \beta(s))$:*

$$(Q(s,a) - V(s)) \sim -\mathcal{G}(0, \beta(s)).$$

**Remark on Assumption 1** This assumption is identical to the Gumbel error model used in Double Gumbel Q-learning (Hui et al., 2023), where the learned $Q$-function is modeled as $Q^\star$ perturbed by Gumbel-distributed errors arising from the max-operator in the Bellman equation. It provides an analytically tractable characterization of the heteroscedastic, state-dependent approximation error in deep Q-learning, which is more realistic than homoscedastic Gaussian noise.

**Remark on Assumption 2** Assumption 2 is similar to the one used in XQL (Garg et al., 2023), except that it allows for a state-dependent noise model. It reduces exactly to the XQL assumption when the noise scale is constant, i.e., $\beta(s) = \beta$.

With the above assumptions in place, we now present Proposition 1, followed by Proposition 2, which provides key theoretical support for estimating $\beta(s)$.

**Proposition 1.** *Under Assumptions 1 and 2, given an optimal $Q$-function $Q^\star$ and $V^\star(s) = \max_a Q^\star(s,a)$, the following equation holds:*

$$V^\star(s) = \beta(s) \log \int \exp \left( \frac{Q^\star(s,a)}{\beta(s)} \right) da.$$

*Proof.* See Appendix B.1. $\square$

Proposition 1 establishes a relationship between the optimal $Q$-function and value function. To extend this relationship to the estimated functions used in practice, we define an estimated value function inspired by the structure in Proposition 1.

**Definition 1.** *Given a Q-function estimation and a state-dependent temperature parameter $\beta(s)$, we define the following value function:*

$$\tilde{V}(s) = \beta(s) \log \int \exp\left(\frac{Q(s,a)}{\beta(s)}\right) da.$$

**Remark on Definition 1** $\tilde{V}(s)$ can be interpreted as a SoftArgMax (Hui et al., 2023) of $Q(s,a)$, aligning with the principles of MaxEnt RL and Extreme Q-Learning. Since we assume the $\beta(s)$ scale is consistent across Assumptions 1 and 2, it enables a coherent formulation of both $V(s)$ and $\tilde{V}(s)$ (Definition 1). For simplicity, we use the notation $V(s)$ throughout the paper. Empirical evidence supporting the consistency of the $\beta(s)$ scale is presented in the experimental section.

Based on Proposition 1 and Definition 1, we can construct an estimation of $\beta(s)$ based on the value functions:

**Proposition 2** (Estimating $\beta(s)$). *Under Assumptions 1 and 2, given $\hat{V}(s) = \mathbb{E}[V^\star(s)]$, there exists:*

$$\omega\beta(s) = \hat{V}(s) - V(s),$$

*where $\omega \approx 0.57721$ is identified as the Euler–Mascheroni constant.*

*Proof.* See Appendix B.2. $\qquad\square$

In this case, $\beta(s)$ can be estimated by computing $[\hat{V}(s) - V(s)]/\omega$. However, how to estimate both $\hat{V}(s)$ and $V(s)$ remains an open question. In the next section, we propose an estimation method based on quantile regression.

## 4.2 Learning Value Functions with Quantile Regression

In this section, we demonstrate how the learning of $V(s)$ and $\hat{V}(s)$ can be formulated as a quantile regression problem. Under Assumption 2, the cumulative distribution function (CDF) of the Gumbel distribution enables us to interpret the value functions as quantiles, as established in Propositions 3 and 4.

**Proposition 3.** *For a Q-function and value function satisfying Assumption 2, and with $\alpha_1 = 1 - \exp(-1)$, there exists:*

$$P(Q(s,a) \leq V(s)) = \alpha_1,$$

*which shows that $V(s)$ corresponds to the $\alpha_1$-quantile of $Q(s,a)$ under its CDF.*

*Proof.* See Appendix B.3. $\qquad\square$

Similarly, by applying the results from Proposition 2, we obtain the following for $\hat{V}$.

**Proposition 4.** *Under Assumptions 1 and 2, for a Q-function, let $\hat{V}(s) = \mathbb{E}[V^\star(s)]$, and define $\alpha_2 = 1 - \exp(-\exp(\omega))$. Then, there exists:*

$$P(Q(s,a) \leq \hat{V}(s)) = \alpha_2,$$

*which shows that $\hat{V}(s)$ corresponds to the $\alpha_2$-quantile of $Q(s,a)$ under its CDF.*

*Proof.* See Appendix B.4. $\qquad\square$

Based on the preceding propositions, we have established that $V(s)$ and $\hat{V}(s)$ correspond to the $\alpha_1$- and $\alpha_2$-quantiles of $Q(s,a)$ under its cumulative distribution function. Consequently, it is natural to estimate $V(s)$ and $\hat{V}(s)$ using quantile regression, and to estimate $\beta(s)$ based on the resulting estimates of $V(s)$ and $\hat{V}(s)$.

---

**Algorithm 1** Quantile $Q$-Learning

---

**Require:** Initialized networks $Q_\theta$, $V_{\psi_1}$, $V_{\psi_2}$, dataset $\mathcal{D}$, mild generalization and policy constraint hyperparameters $\lambda$ and $\zeta$

1: **for** each gradient step **do**
2:     Sample a batch of transitions from dataset $\mathcal{D}$
3:     Update value networks by minimizing Eqs. 7.
4:     Update $Q$-function $Q_\theta$ by minimizing Eq. 5
5:     Update policy $\pi_\phi$ by maximizing Eq. 6
6: **end for**

---

**Definition 2** (Quantile Regression). *The quantile regression objective is defined as:*

$$\mathcal{L}_{qr}(u, \tau) = u \cdot (\tau - \mathbb{I}(u < 0)),$$

*where $u = y - \hat{y}$ is the residual, $\tau \in [0, 1]$ is the target quantile level, and $\mathbb{I}(\cdot)$ denotes the indicator function.*

To learn $V(s)$ and $\hat{V}(s)$, we parameterize them using $\psi_1$ and $\psi_2$, respectively. Leveraging the quantile regression objective defined in Definition 2, we formulate the learning objectives for $V_{\psi_1}(s)$ and $\hat{V}_{\psi_2}(s)$ as follows:

$$\mathcal{J}(\psi_1) = \mathbb{E}_{(s,a)\sim\mathcal{D}} \, \mathcal{L}_{qr}(Q_\theta(s, a) - V_{\psi_1}(s), \alpha_1). \tag{3}$$

$$\mathcal{J}(\psi_2) = \mathbb{E}_{(s,a)\sim\mathcal{D}} \, \mathcal{L}_{qr}(Q_\theta(s, a) - \hat{V}_{\psi_2}(s), \alpha_2). \tag{4}$$

For a $Q$-function satisfying Assumption 1, $Q(s, a) + \omega\beta(s)$ is an unbiased estimation of the optimal $Q$-function $Q^\star(s, a)$. By Proposition 2, the estimator for $Q^\star$ can be further expressed as $Q(s, a) + (\hat{V}(s) - V(s))$. Therefore, the $Q$-function, parameterized by $\theta$, is learned using the mean-squared Bellman error :

$$\mathcal{L}_Q(\theta) = \mathbb{E}_{(s,a,s')\sim\mathcal{D}} \left[ Q_\theta(s, a) + (\hat{V}_{\psi_2}(s) - V_{\psi_1}(s)) - r(s, a) - \gamma\hat{V}_{\psi_2}(s') \right]^2. \tag{5}$$

Using the estimator $\beta(s) = (\hat{V}_{\psi_2}(s) - V_{\psi_1}(s))/\omega$ and a KL-constrained policy objective, we can obtain a $\beta$-free AWR-style policy objective with a policy constraint weight $\zeta$:

$$\mathcal{L}_\pi(\phi) = \mathbb{E}_{(s,a)\sim\mathcal{D}} \exp\left( \frac{\omega(Q_\theta(s, a) - V_{\psi_2}(s))}{\zeta(\hat{V}_{\psi_2}(s) - V_{\psi_1}(s))} + \frac{\omega(Q_\theta(s, a) - V_{\psi_1}(s))}{\hat{V}_{\psi_2}(s) - V_{\psi_1}(s)} \right) \log \pi_\phi(s, a). \tag{6}$$

The detailed derivation can be found in Appendix A. Thus far, we have established the basic formulation of our proposed algorithm, a $\beta$-free variant of XQL, which we refer to as **Quantile $Q$-Learning**. Further enhancements to this approach will be presented in the following sections.

### 4.3 Regulating Value Functions with Mild Generalization

By estimating $\beta$ via quantile regression, we construct a $\beta$-free variant of XQL. However, similarly to the original XQL, this approach remains fully in-sample and has been shown to be overly conservative (Mao et al., 2024). To address this crucial issue, we introduce mild generalization into the quantile regression objective.

A straightforward approach is to apply Doubly Mild Generalization (DMG) (Mao et al., 2024) directly to Eq.3 and Eq.4. However, approximating $V^\star(s') = \max_{a'\sim\pi(\cdot|s')} Q^\star(s', a')$ by sampling actions from $\pi(\cdot \mid s')$ and taking the $\alpha_2$-quantile of $Q_\theta(s', a')$ can introduce extrapolation error (Fujimoto et al., 2019), which may propagate throughout the learning process. To mitigate this, we conservatively estimate $V^\star(s')$ by subtracting $\omega\beta(s')$, with $\beta(s')$ providing an estimation of uncertainty. Applying this conservative estimation systematically to all value functions throughout mild generalization leads to $V(s')$ as the $\alpha_0$-quantile of $Q(s', a')$, and $\hat{V}(s)$ as the $\alpha_1$-quantile of $Q(s', a')$, whereby $\alpha_0 = 1 - \exp(-\exp(-\omega))$. Ablation results in the experiment section demonstrate the effectiveness of this modification. The updated objective for training the value networks incorporating mild generalization can therefore be expressed as:

$$\begin{aligned}
\mathcal{J}(\psi_1) &= \mathbb{E}_{(s,a)\sim\mathcal{D}} \, \mathcal{L}_{qr}(Q_\theta(s, a) - V_{\psi_1}(s), \alpha_1) + \mathbb{E}_{s'\sim\mathcal{D}, a'\sim\pi(\cdot|s')} \lambda \, \mathcal{L}_{qr}(Q_\theta(s', a') - V_{\psi_1}(s'), \alpha_0), \\
\mathcal{J}(\psi_2) &= \mathbb{E}_{(s,a)\sim\mathcal{D}} \, \mathcal{L}_{qr}(Q_\theta(s, a) - \hat{V}_{\psi_2}(s), \alpha_2) + \mathbb{E}_{s'\sim\mathcal{D}, a'\sim\pi(\cdot|s')} \lambda \, \mathcal{L}_{qr}(Q_\theta(s', a') - \hat{V}_{\psi_2}(s'), \alpha_1).
\end{aligned} \tag{7}$$

In this objective, a mild generalization hyperparameter $\lambda$ is introduced to control the degree of generalization applied. Empirically, we find that $\lambda = 1$ works well in most cases.

## 5 Experiments

### 5.1 Experimental Setting and Baseline Methods

**Benchmark Datasets**   We conduct experiments on two offline RL benchmarks: the widely used D4RL suite (Fu et al., 2020) and the more challenging, near–real-world NeoRL2 benchmark (Gao et al., 2025), which incorporates real-world complexities such as high-latency transitions and global safety constraints. For D4RL, our evaluation spans locomotion tasks (Hopper, HalfCheetah, and Walker2d) across various dataset types, including medium, medium-replay, and medium-expert. We also evaluate on the Adroit manipulation tasks (Pen, Door, and Hammer), as well as the AntMaze-Umaze task under both fixed and randomized initial states and goals. For NeoRL2, we assess performance on the RocketRecovery and SafetyHalfCheetah tasks.

**Baseline Methods**   Our approach is compared against a diverse set of offline RL baselines, encompassing various design principles. These include policy-constrained methods such as Behavior Cloning (BC) and TD3+BC (Fujimoto & Gu, 2021); conservative model-free algorithms like CQL (Kumar et al., 2020); in-sample learning methods including IQL (Kostrikov et al., 2021) and XQL (Garg et al., 2023); a stabilized variant of XQL, MXQL (Omura et al., 2024); an uncertainty-aware method, EDAC (An et al., 2021); and the batch-constrained off-policy algorithm BCQ (Fujimoto et al., 2019).

**Hyperparameter Setting**   For every task–dataset combination we fix the generalization coefficient to $\lambda = 1.0$ and the policy constraint weight to $\zeta = 1.0$. Using this single hyperparameter setting for all of our experiments (Table 1) highlights the robustness of our method, whereas baseline algorithms are evaluated with the dataset-specific, tuned hyperparameters reported in prior work. We also compare our approach and XQL under a consistent hyperparameter setting across tasks, revealing that the original XQL algorithm suffers from performance degradation without per-task tuning, while our method maintains strong performance (Table 3). The detailed hyperparameter settings for the baseline algorithms are provided in Appendix F.

### 5.2 Main Results

In this section, the results of the proposed algorithm are presented on D4RL datasets, and its performance is compared with baseline methods. Our approach consistently outperforms or matches state-of-the-art in-sample and model-free offline RL baselines under dataset-specific hyperparameter tuning using a unified hyperparameter setting across diverse domains (Locomotion, Adroit, and AntMaze) and dataset types. The results are summarized in Table 1. Furthermore, training curve comparisons are presented between two improved versions of XQL: MXQL and our proposed QQL. As shown in Figure 2a, QQL demonstrates more stable training dynamics, with lower standard deviation, and achieves more optimal performance. In terms of computational cost, QQL requires approximately $1.6\times$ the wall-clock training time of XQL (4 hours vs. 2.5 hours for 1M timesteps on an RTX 4070), primarily due to the additional quantile regression and value estimation. We consider this overhead acceptable given the gains in stability and final performance.

### 5.3 Comparison with Consistent Hyperparameter Setting

We compare the performance of our method with XQL (Garg et al., 2023) under a consistent hyperparameter setting across diverse datasets and domains. As shown in Table 3, XQL suffers significant performance degradation when using the consistent hyperparameters across settings, while our method maintains strong performance. This highlights the robustness and generality of our approach.

| Domain | Dataset | BC | TD3+BC | CQL | IQL | XQL | MXQL | QQL (Ours) |
|---|---|---|---|---|---|---|---|---|
| | hopper-med-exp | 52.5 | 98.0 | 105.4 | 91.5 | 111.2 | 110.7 | **112.5±1.3** |
| | hopper-med | 52.9 | 59.3 | 58.5 | 66.3 | 74.2 | **80.9** | 77.3 ± 3.8 |
| | hopper-med-rep | 18.1 | 60.9 | 95.0 | 94.7 | 100.7 | **102.7** | 101.1±2.1 |
| | halfcheetah-med-exp | 55.2 | 90.7 | 91.6 | 86.7 | 94.2 | 92.1 | **94.3 ± 1.8** |
| Gym Locomotion | halfcheetah-med | 42.6 | 48.3 | 44.0 | 47.4 | 48.3 | 47.7 | **49.5 ± 0.3** |
| | halfcheetah-med-rep | 36.6 | 44.6 | 45.5 | 44.2 | 45.2 | 45.7 | **46.6 ± 0.3** |
| | walker2d-med-exp | 107.5 | 110.1 | 108.8 | 112.7 | 112.7 | 111.2 | **113.2±0.3** |
| | walker2d-med | 75.3 | 83.7 | 72.5 | 78.3 | 84.2 | 83.8 | **85.2 ±1.3** |
| | walker2d-med-rep | 26.0 | 81.8 | 77.2 | 73.9 | 82.2 | 83.6 | **90.2 ± 2.1** |
| | pen-human | 99.7 | 10.0 | 58.9 | 106.2 | 105.3 | 122.1 | **128.3 ± 4.1** |
| | pen-cloned | 99.1 | 52.7 | 14.7 | 114.1 | 112.6 | **117.4** | 115.2 ± 4.7 |
| Adroit | door-human | 9.4 | -0.1 | 13.3 | 13.5 | 13.2 | 18.3 | **21.1 ± 5.6** |
| | door-cloned | 3.4 | -0.2 | -0.1 | **9.0** | 1.1 | 1.8 | 8.8 ± 4.1 |
| | hammer-human | 12.6 | 2.4 | 0.3 | 6.9 | 7.3 | **14.7** | 8.4 ± 3.1 |
| | hammer-cloned | 8.9 | 0.9 | 0.3 | **11.6** | 1.1 | 11.1 | 8.0 ± 4.4 |
| AntMaze | antmaze-umaze | 68.5 | **98.5** | 94.8 | 84.0 | 90.3 | 88.3 | 88.5 ± 6.1 |
| | antmaze-umaze-diverse | 64.8 | 71.3 | 53.8 | 79.5 | 77.2 | 53.2 | **81.3 ± 4.1** |

Table 1: **Main Experimental Results on D4RL.** Average normalized performance on Gym Locomotion, Adroit, and AntMaze tasks. All hyperparameters for our QQL method are kept **consistent** across all environments and datasets, while baseline algorithms are individually **tuned** for each setting. Results are averaged over 5 random seeds.

| Dataset | DATA | BC | CQL | EDAC | BCQ | TD3+BC | XQL | QQL (Ours) |
|---|---|---|---|---|---|---|---|---|
| RocketRecovery | 75.27 | 72.75 | 74.32 | 65.65 | 76.46 | 79.74 | 81.1 | **83.2±4.7** |
| SafetyHalfCheetah | **73.56** | 70.16 | 71.18 | 53.11 | 54.65 | 68.58 | 66.2 | 60.3±4.1 |

Table 2: **Main Experimental Results on NeoRL2.** Average normalized performance on RocketRecovery and SafetyHalfCheetah tasks. All hyperparameters for our QQL method are kept consistent, while baseline algorithms are individually **tuned** for each setting. The **DATA** column reports the normalized scores of the trajectories contained in the offline datasets. Results are averaged over 3 random seeds.

## 5.4 Ablation Studies

**Ablations on Value Regulation and Conservative Estimation** Ablation studies are performed to isolate the contributions of Value Regulation (VR) and Conservative Estimation (CE) to the performance and stability of the QQL algorithm. Experiments on HalfCheetah Medium, Walker2d Medium Replay, and Hopper Medium Expert (Figure 3) compare the full QQL method with variants that exclude value regulation (QQL w/o VR) or conservative estimation (QQL w/o CE), as described in the methodology section.

Removing either component results in noticeable performance drops, while omitting both leads to substantial degradation. This highlights the role of value regulation in mitigating overestimation and stabilizing value estimates for robust offline learning, and the importance of conservative estimation in encouraging pessimistic value functions to better handle distributional shifts and limited coverage.

These results demonstrate that both VR and CE are essential for effective offline RL. Their complementary effects yield a more stable and performant learning algorithm. All ablation experiments use consistent hyperparameter settings.

Additionally, a comparison of the Q-values from the original QQL method against those from QQL without value regularization and without conservative estimation was plotted. The results demonstrate that both value regularization and conservative estimation help stabilize Q-value estimation and improve conservative-

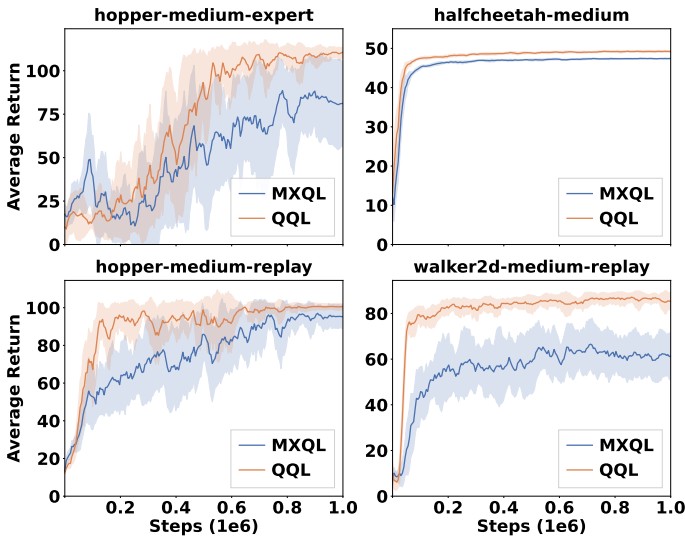
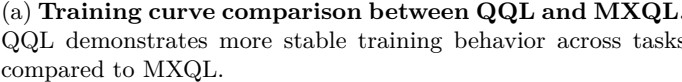
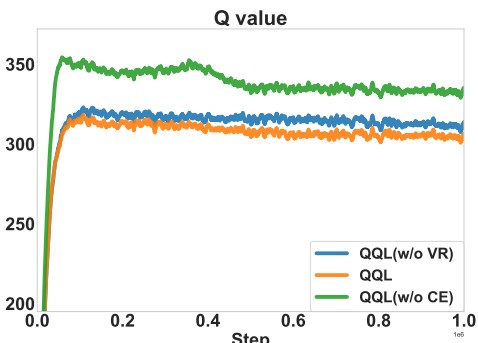

(a) **Training curve comparison between QQL and MXQL.** QQL demonstrates more stable training behavior across tasks compared to MXQL.

(b) **Q-value plots for value regularization and conservative estimation ablation.** We visualize the Q-value estimates of the original QQL method, its variant without value regularization (w/o VR), and its variant without conservative estimation (w/o CE), as described in the methodology section. The ablation results demonstrate that the proposed approach effectively stabilizes Q-value estimation.

Figure 2: **QQL Performance and Ablation Analysis.** We visualize the following results: (a) Training stability comparison between QQL and MXQL. (b) Q-value plots demonstrating the stabilization effect of value regularization and conservative estimation.

| Domain | Gym Locomotion | Adroit | AntMaze |
|---|---|---|---|
| XQL (Dataset-specific tuning) | 83.7±2.7 | 40.6±6.3 | 83.8±6.9 |
| XQL (Consistent per domain) | 75.5±3.6 | 38.6±4.5 | 69.6±10.7 |
| XQL (Consistent across all domains) | 73.6±4.1 | 37.8±4.1 | 67.3±9.2 |
| QQL (Consistent across all domains) | **85.6±1.4** | **48.3±4.3** | **84.9±5.1** |

Table 3: **Comparisons with Consistent Hyperparameters** We compare our approach—using a single, consistent set of hyperparameters—against three XQL variants: (1) with dataset-specific tuning, (2) with domain-level consistent hyperparameters, and (3) with a single set shared across all domains. Performance is reported as the average D4RL score across all dataset types and tasks within each domain. Detailed hyperparameter settings for the different XQL variants are provided in Appendix F.1. All results are averaged over 5 random seeds.

ness. The ablation study is performed on the Hopper Medium Expert dataset, with the results shown in Figure 2b.

**Ablations on the Hyperparameters $\lambda$ and $\zeta$**  Ablation studies are conducted to investigate the impact of the mild generalization coefficient $\lambda$ and the policy constraint weight $\zeta$. Experiments are performed on three Gym Locomotion tasks: HalfCheetah Medium, Walker2d Medium Replay, and Hopper Medium Expert. For both $\lambda$ and $\zeta$, we fix one hyperparameter at 1.0 while varying the other over the set $[0.25, 0.5, 1.0, 2.0, 4.0]$. The results are summarized in Table 4.

The ablation results show that the scores exhibit minimal variation with changes in the hyperparameters $\lambda$ and $\zeta$, demonstrating the robustness of the proposed QQL method to these hyperparameters. The ablation results show that the scores exhibit minimal variation with changes in the hyperparameters $\lambda$ and $\zeta$, demonstrating the robustness of the proposed QQL method to these hyperparameters.

| Dataset | Setting | 0.25 | 0.5 | 1.0 | 2.0 | 4.0 |
|---------|---------|------|-----|-----|-----|-----|
| hopper-med-exp | $\lambda = 1.0$ ($\zeta$ varies) | 113.2±0.3 | 113.1±0.3 | 112.5±1.3 | 112.9±0.3 | 112.6±0.6 |
|  | $\zeta = 1.0$ ($\lambda$ varies) | 110.7±3.2 | 111.6±2.1 | 112.5±1.3 | 112.1±1.7 | 112.6±0.3 |
| halfcheetah-med | $\lambda = 1.0$ ($\zeta$ varies) | 50.0±0.1 | 49.6±0.1 | 49.5±0.3 | 49.4±0.1 | 49.1 ± 0.1 |
|  | $\zeta = 1.0$ ($\lambda$ varies) | 49.6±0.1 | 49.6±0.1 | 49.5±0.3 | 49.6±0.1 | 49.6±0.2 |
| walker2d-med-rep | $\lambda = 1.0$ ($\zeta$ varies) | 90.9±0.8 | 90.4±1.2 | 90.2±2.1 | 89.9±1.5 | 88.4±0.4 |
|  | $\zeta = 1.0$ ($\lambda$ varies) | 88.8 ± 1.4 | 90.5 ± 1.0 | 90.2±2.1 | 88.5 ± 0.6 | 89.3 ± 0.6 |

Table 4: **Ablations on $\lambda$ and $\zeta$.** Average normalized performance on D4RL tasks. The first row for each task shows performance when varying $\zeta$ with fixed $\lambda = 1.0$, and the second row shows performance when varying $\lambda$ with fixed $\zeta = 1.0$.

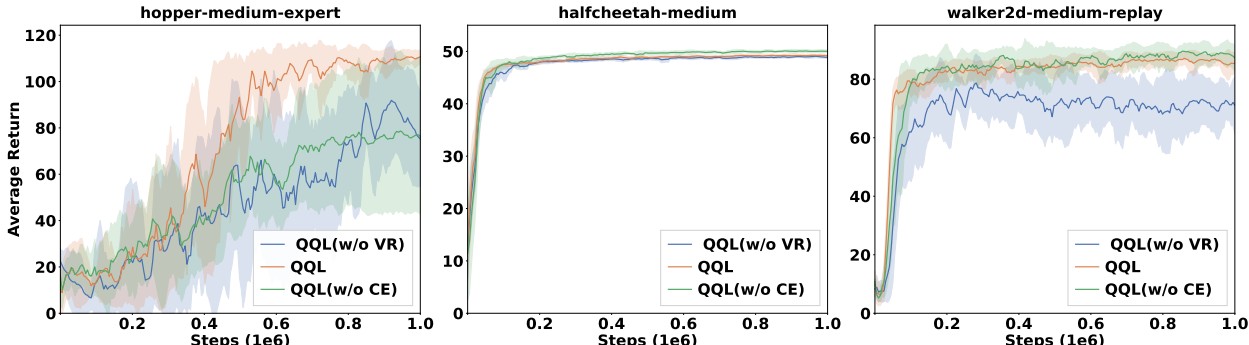

Figure 3: **Ablation Studies on Value Regulation and Conservative Estimation** Comparison of QQL performance to its variant without value regulation (w/o VR) and its variant without conservative estimation (w/o CE) adapted in the methodology section. The hyperparameters are consistent across variants.

## 5.5 Experiments on $\beta(s)$ Scale

In Definition 1, we actually assume that the $\beta(s)$ scale of Assumption 1 and Assumption 2 is similar, as discussed in the remark of Definition 1. In this section, we provide empirical evidence by leveraging the results from a toy example. Additional empirical support, including a residual analysis on D4RL tasks, is provided in Appendix H.

**Toy Example Construction.** To illustrate that Assumption 2 can be approximately satisfied given Assumption 1, we construct a toy example where the Q-value estimation process aligns with the desired Gumbel properties. Under Assumption 1, we model $-Q(s, a)$ as a Gumbel-distributed random variable with location $-Q^\star(s, a)$ and scale $\beta(s)$. We fix the state $s$ and generate $-Q^\star(s, a)$ using a randomly initialized neural network that takes action $a$ as input. To simulate $-Q(s, a)$, we add Gumbel noise $g(s) \sim \mathcal{G}(0, \beta(s))$ to the output of the network. To test Assumption 2, we sample actions from a Gaussian policy and analyze the resulting distribution of $-Q(s, a)$ values to check whether it retains the Gumbel form. The procedure involves computing $-Q^\star(s, a)$ for each sampled action, adding Gumbel noise, and comparing the empirical distribution of $-Q(s, a)$ to a theoretical Gumbel distribution. We fix the Gumbel noise scale at $\beta(s) = 1$ and vary the standard deviation of the Gaussian policy over a range of values.

**Results and Discussion.** Figure 4 displays the empirical distributions of $-Q(s, a)$ under varying Gaussian policy variances, overlaid with Gumbel distributions fitted via maximum likelihood estimation. The strong alignment between the empirical histograms and the fitted Gumbel curves, which is confirmed with high p-values in a goodness-of-fit test, shows that the distribution of $-Q(s, a)$ maintains its Gumbel form despite the stochasticity of Gaussian sampling. The shape and scale of these fitted distributions are primarily controlled by the additive Gumbel noise used in the simulation. This supports our premise that the scale parameter

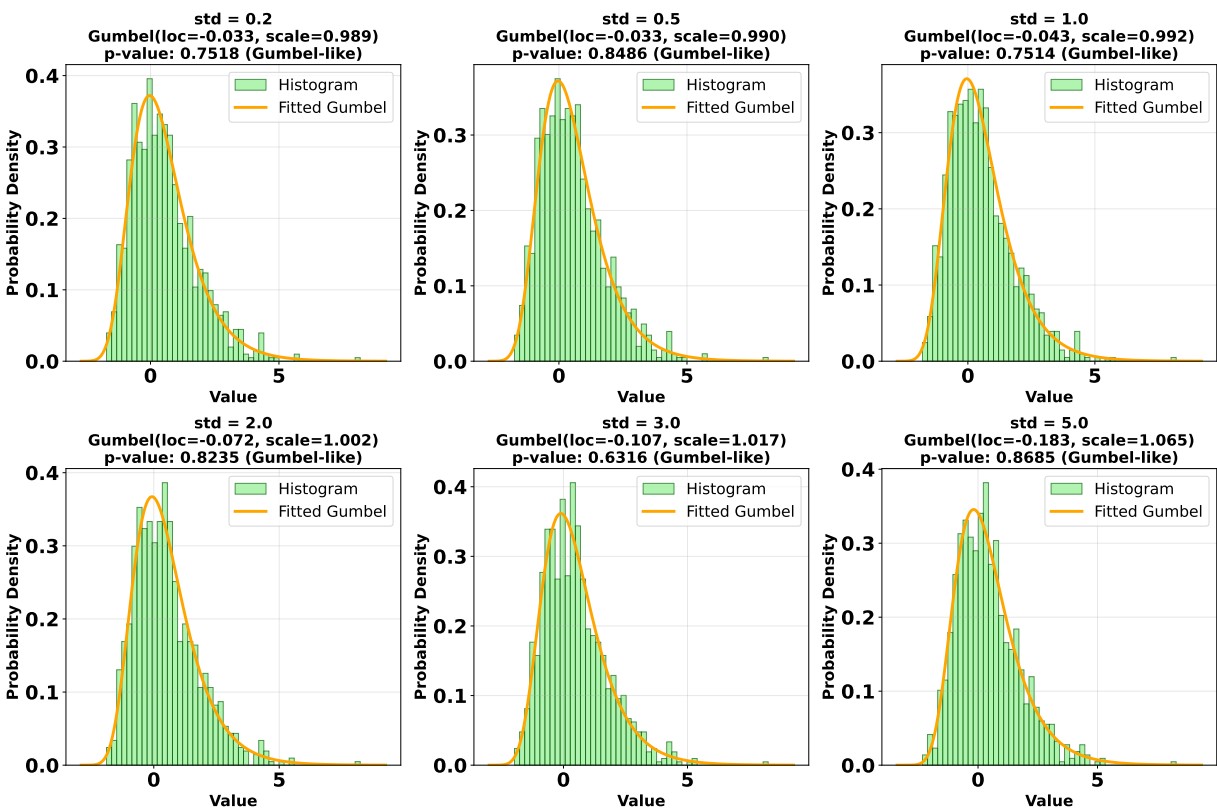

Figure 4: **Toy Example for** $\beta(s)$ **Scale.** Empirical distributions of $-Q(s, a)$, along with the corresponding fitted Gumbel distributions and their estimated location, scale, and goodness-of-fit p-values.

$\beta(s)$ stays consistent between Assumptions 1 and 2, even when actions are sampled from a distribution rather than being deterministically fixed.

## 6 Conclusion and Future Work

In this work, we have addressed critical limitations of prior Extreme $Q$-Learning (XQL) approaches—namely, the need for dataset-specific hyperparameter tuning and the instability of training. We have introduced a novel method for estimating the temperature coefficient $\beta$ via quantile regression, requiring only mild statistical assumptions. To further bolster stability, we have incorporated a value regularization mechanism inspired by constrained value learning that encourages mild generalization while preserving performance.

Empirical results across diverse offline RL benchmarks, including D4RL and NeoRL2, validate our approach: it achieves performance that is competitive with—or even surpasses—XQL and its stabilized variant MXQL, while avoiding the hyperparameter overfitting and erratic training behavior they exhibit. Crucially, our method maintains robust performance using a consistent set of hyperparameters across all tasks and domains, highlighting its practicality and general applicability in real-world high-stakes settings.

Looking ahead, a natural extension of this work is to adapt our approach to the online reinforcement learning setting by leveraging developments in online variants of Extreme $Q$-Learning. Integrating our quantile-based temperature estimation and value regularization into the online learning framework could improve training stability in interactive environments. This generalization has the potential to make Extreme $Q$-Learning more robust and widely applicable in real-time decision-making scenarios.

## Acknowledgments

This work was supported by the National Natural Science Foundation of China under Grant No. 62233004, Jiangsu Provincial Scientific Research Center of Applied Mathematics under Grant No. BK20233002, Jiangsu Funding Program for Excellent Postdoctoral Talent under Grant No. 2025ZB481, and China Postdoctoral Science Foundation under Grant No. 2025M771682.

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

## A    Derivation of Policy Objective

A common challenge in policy learning arises when the sampling policy $\mu(\cdot \mid \cdot)$ is suboptimal, which often results in overly conservative estimates from in-sample approaches. To address this issue, existing methods like Advantage-Weighted Regression (AWR) aim to mitigate such inherent conservativeness. Specifically, the AWR-style policy objective minimizes the cross-entropy loss $\mathbb{C}(\mu^* \parallel \pi)$, where $\mu^*(a \mid s) \propto \mu(a \mid s) \exp(Q(s,a)/\beta(s))$. We define the normalizer $V(s) = \beta(s) \int \mu(a \mid s) \exp(\frac{Q(s,a)}{\beta(s)}) da$. Thus, $\mu^*$ can be expressed as $\mu^*(a \mid s) = \mu(a \mid s) \exp(\frac{Q(s,a) - V(s)}{\beta(s)})$.

Given that $\mu^*$ shares the same support as $\mu$ and is considered a superior policy, we construct a less conservative policy $\pi$ by remaining close to $\mu^*$. Consequently, the following objective is maximized:

$$\max_{\phi} \mathbb{E}_{s\sim\mathcal{D}, a\sim\pi_\phi(\cdot|s)} Q_\theta(s,a) - \nu \mathbb{E}_{(s,a)\sim\mathcal{D}} \mathbf{KL}(\mu^* \parallel \pi_\phi(\cdot \mid s)). \tag{8}$$

This objective is equivalent to:

$$\max_{\phi} \mathbb{E}_{s\sim\mathcal{D}, a\sim\pi_\phi(\cdot|s)} Q_\theta(s,a) - \nu \mathbb{E}_{(s,a)\sim\mathcal{D}} [\exp((Q_\theta(s,a) - V_{\psi_1})/\beta(s)) \log \pi_\phi(a \mid s)]. \tag{9}$$

Here, $\nu = \zeta\beta(s)$, where $\zeta$ is a positive, adjustable parameter referred to as the policy constraint weight, balancing policy optimization and the conservative constraint.

Furthermore, Eq. 9 resembles the MaxEnt RL objective and possesses a closed-form solution:

$$
\begin{aligned}
\pi^*(a \mid s) &= \mu^*(a \mid s) \exp\left(\frac{Q(s,a) - V'(s)}{\beta(s)}\right) \\
&= \mu(a \mid s) \exp\left(\frac{Q(s,a) - V'(s)}{\zeta\beta(s)}\right) \exp\left(\frac{Q(s,a) - V(s)}{\beta(s)}\right),
\end{aligned}
$$

where $V'(s) = \beta(s) \int \mu^*(a \mid s) \exp(\frac{Q(s,a)}{\zeta\beta(s)}) da$ serves as a normalizer. Based on this, the policy optimization objective is modified to minimize the KL divergence between $\pi^*$ and $\pi$. In practice, $V_{\psi_1}$ and $V_{\psi_2}$ approximate the normalizers $V(s)$ and $V'(s)$, respectively, leading to the following policy optimization objective:

$$\max_\phi \mathbb{E}_{(s,a)\sim\mathcal{D}} \exp \left( \frac{Q_\theta(s,a) - V_{\psi_2}(s)}{\zeta((V_{\psi 2}(s) - V_{\psi 1}(s))/\omega)} + \frac{Q_\theta(s,a) - V_{\psi_1}(s)}{(V_{\psi 2}(s) - V_{\psi 1}(s))/\omega} \right) \log \pi_\phi(s,a).$$

## B  Omitted Proofs

### B.1  Proof for Proposition 1

To begin with, we introduce the following lemmas:

**Lemma 1** (Gumbel 1935). *Given a continuous random variable $\alpha$ and a Gumbel noise $g \sim \mathcal{G}(0, \beta)$, the following identity exists:*

$$\beta \log \int \exp \left( \frac{\alpha + g}{\beta} \right) d\alpha = \beta \log \int \exp \left( \frac{\alpha}{\beta} \right) d\alpha + g.$$

*Proof.*

$$\begin{aligned}
\beta \log \int \exp \left( \frac{\alpha + g}{\beta} \right) d\alpha &= \beta \log \int \exp \left( \frac{\alpha}{\beta} + \frac{g}{\beta} \right) d\alpha \\
&= \beta \log(\exp \left( \frac{g}{\beta} \right) \int \exp \left( \frac{\alpha}{\beta} \right) d\alpha) \\
&= \beta \log \exp \left( \frac{g}{\beta} \right) + \beta \log \int \exp \left( \frac{\alpha}{\beta} \right) d\alpha \\
&= \beta \log \int \exp \left( \frac{\alpha}{\beta} \right) d\alpha + g.
\end{aligned}$$

$\square$

**Lemma 2** (Hui et al. 2023). *Given a optimal Q-function $Q^\star$ and a state-dependent gumbel noise $g(s) \sim \mathcal{G}(0, \beta(s))$, the following equation holds:*

$$\max_a Q^\star(s,a) = \beta(s) \log \int \exp \left( \frac{Q(s,a)}{\beta(s)} \right) da + g(s). \tag{10}$$

*Proof.* See prior work (Hui et al., 2023). $\square$

With the lemmas above, we're ready to provide the proof for Proposition 1:

*Proof.* By definition, $V^\star(s) = \max_a Q^\star(s,a)$, and according to Lemma 2, we have:

$$V^\star(s) = \beta(s) \log \int \exp \left( \frac{Q(s,a)}{\beta(s)} \right) da + g(s).$$

And by Lemma 1, it exists:

$$\begin{aligned}
V^\star(s) &= \beta(s) \log \int \exp \left( \frac{Q(s,a) + g(s,a)}{\beta(s)} \right) da \\
&= \beta(s) \log \int \exp \left( \frac{Q^\star(s,a)}{\beta(s)} \right) da.
\end{aligned}$$

The last part is completed by directly applying Assumption 1. $\square$

### B.2 Proof for Proposition 2

Given $\hat{V}(s) = \mathbb{E}[V^\star(s)]$, Proposition 1 and Assumption 1, we can construct the proof for Proposition 2:

*Proof.*

$$\hat{V}(s) - V(s) = \mathbb{E}\left[\beta(s)\log\int\exp\left(\frac{Q^\star(s,a)}{\beta(s)}\right)da - \beta(s)\log\int\exp\left(\frac{Q^\star(s,a) - g(s)}{\beta(s)}\right)da\right]$$

$$= \mathbb{E}\beta(s)\left[\log\int\exp\left(\frac{Q^\star(s,a)}{\beta(s)}\right)da - \log\int\exp\left(\frac{Q^\star(s,a)}{\beta(s)}\right)da + \frac{g(s)}{\beta(s)}\right]$$

$$= \mathbb{E}[g(s)] = \omega\beta(s).$$

The final equation leverages a property of the Gumbel distribution to compute the expectation:

$$\mathbb{E}[g] = \omega\beta, \quad \text{where} \quad g \sim \mathcal{G}(0, \beta).$$

$\square$

### B.3 Proof for Proposition 3

Leveraging Assumption 2 and the properties of the Gumbel distribution, we can construct the following proof:

*Proof.*
$$P(Q(s,a) \le V(s)) = 1 - P((Q(s,a) > V(s))$$
$$= 1 - P(-g(s) > 0)$$
$$= 1 - P(g(s) < 0)$$
$$= 1 - \exp(-1) = \alpha_1,$$

where we denote $g(s) \sim \mathcal{G}(0, \beta(s))$.

$\square$

### B.4 Proof for Proposition 4

Similar to the proof of Proposition 3, the proof of Proposition 4 can be constructed using Assumption 2, Proposition 2, and the properties of the Gumbel distribution:

*Proof.*
$$P(Q(s,a) \le \hat{V}(s)) = 1 - P((Q(s,a) > \hat{V}(s))$$
$$= 1 - P(-g(s) - \omega\beta(s) > 0)$$
$$= 1 - P(g(s) < -\omega\beta(s))$$
$$= 1 - \exp(-\exp(\omega)),$$

where we denote $g(s) \sim \mathcal{G}(0, \beta(s))$. $\square$

## C   Analysis of the Error Model

For completeness, this section provides a formal exposition of the Rust–McFadden model and the Gumbel Error Model (GEM), both of which underpin the theoretical foundation of extreme value theory in reinforcement learning. For a more comprehensive treatment, we refer the reader to (Garg et al., 2023; McFadden, 1972).

### C.1 Rust–McFadden Model for Markov Decision Processes

We consider a Markov decision process (MDP) in which the observed stochasticity in rewards arises not from intrinsic environmental randomness, but from an unobserved latent variable. Formally, the complete state is modeled as a tuple $(s, z)$, where $s \in \mathcal{S}$ denotes the observable state and $z$ represents a latent component that governs reward uncertainty. The associated state–action value and value functions are defined as:

$$Q(s, z, a) = R(s, z, a) + \gamma \, \mathbb{E}_{s' \sim P(\cdot | s, a)} \left[ \mathbb{E}_{z' | s'} [V(s', z')] \right],$$
$$V(s, z) = \max_{a \in \mathcal{A}} Q(s, z, a).$$

The following lemma establishes a critical equivalence between this latent-variable formulation and soft (entropy-regularized) MDPs under standard structural assumptions.

**Lemma 3** (Garg et al. 2023). *Suppose the following conditions hold:*

1. ***Conditional Independence (CI):*** *The latent variable $z'$ depends only on the next observable state $s'$, i.e.,*
$$p(s', z' \mid s, z, a) = p(z' \mid s') \, p(s' \mid s, a).$$

2. ***Additive Separability (AS):*** *The reward function decomposes additively as*
$$R(s, a, z) = r(s, a) + \epsilon(z, a),$$
*where $r(s, a)$ is the deterministic component and $\epsilon(z, a)$ captures latent perturbations.*

*If the perturbations $\epsilon(z, a)$ are independent and identically distributed according to the Gumbel distribution $\mathcal{G}(0, \beta)$, then the optimal value functions satisfy the soft-Bellman equations with entropy regularization parameter $\beta$. Specifically, the state–action value function admits the decomposition $Q(s, z, a) = q(s, a) + \epsilon(z, a)$, and the marginal value function is given by $v(s) = \mathbb{E}_z[V(s, z)]$.*

Consequently, a maximum entropy MDP (MaxEnt MDP) is distributionally equivalent to a standard MDP augmented with i.i.d. Gumbel noise in the reward, provided that the CI and AS conditions are satisfied.

This equivalence is further strengthened by the following converse result.

**Corollary 1** (McFadden 1972). *Consider an MDP that satisfies the Bellman optimality equation and admits a policy of the form $\pi(a \mid s) = \text{softmax}(Q(s, a)/\beta)$. If the randomness in observed rewards arises from i.i.d. latent variables and the AS and CI conditions hold, then these latent variables must follow a Gumbel distribution.*

### C.2 Gumbel Error Model (GEM)

The Gumbel Error Model (GEM) (Garg et al., 2023) provides a distributional perspective on value estimation by modeling Bellman errors as Gumbel-distributed perturbations. This framework elucidates how uncertainty propagates through value iteration and justifies entropy regularization from a statistical standpoint.

#### C.2.1 Model Foundations

Let $\{\hat{Q}_t^{(i)}(s, a)\}_{i=1}^M$ denote $M$ independent estimators of the state–action value function at iteration $t$, and define the expected Q-function as $\bar{Q}_t(s, a) = \mathbb{E}[\hat{Q}_t(s, a)]$. Within the GEM framework, $\bar{Q}_t$ evolves according to the recursion:

$$\bar{Q}_{t+1}(s, a) = r(s, a) + \gamma \, \mathbb{E}_{s' \sim P(\cdot | s, a)} \left[ \mathbb{E}_{\epsilon_t} \left[ \max_{a'} \left( \bar{Q}_t(s', a') + \epsilon_t(s', a') \right) \right] \right], \tag{11}$$

where $\epsilon_t(s', a') \overset{\text{i.i.d.}}{\sim} \mathcal{G}(0, \beta)$ are Gumbel noise terms with scale parameter $\beta > 0$.

A key property of the Gumbel distribution is that the expectation of the perturbed maximum admits a closed-form expression:

$$\mathbb{E}_\epsilon\left[\max_{a'}\left(\bar{Q}(s',a')+\epsilon(s',a')\right)\right] = L^\beta_{a'}[\bar{Q}(s',a')],$$

where $L^\beta_a[Q] := \beta\log\sum_a\exp(Q(s,a)/\beta)$ denotes the log-sum-exp (LSE) operator. Substituting this identity yields the soft-Bellman update:

$$\bar{Q}_{t+1}(s,a) = r(s,a) + \gamma\,\mathbb{E}_{s'\sim P(\cdot|s,a)}\left[L^\beta_{a'}[\bar{Q}_t(s',a')]\right].$$

Under this dynamics, the policy $\pi(a\mid s) = \mathrm{softmax}(\bar{Q}(s,a)/\beta)$ is soft-optimal and maximizes the entropy-regularized expected return.

### C.2.2 Error Dynamics under Deterministic Transitions

In the special case of deterministic dynamics, i.e., $s' = f(s,a)$, the distributional evolution of Q-values can be characterized more precisely. Let $Z_t(s,a) \sim \mathcal{G}(Q_t(s,a),\beta)$ denote a random variable representing the Q-value at time $t$, with independence across state–action pairs. By the Gumbel-max theorem, the maximum over actions satisfies:

$$\max_{a'} Z_t(s',a') \sim \mathcal{G}\left(L^\beta_{a'}[Q_t(s',a')],\beta\right).$$

The Bellman update then takes the form:

$$Z_{t+1}(s,a) = r(s,a) + \gamma\max_{a'} Z_t(s',a').$$

This formulation recovers a standard Bellman operator acting on Gumbel-distributed Q-values, thereby inheriting standard convergence guarantees under mild regularity conditions.

## D   Doubly Mild Generalization

In this section, for completeness, we provide a brief introduction to Doubly Mild Generalization (Mao et al., 2024), which is used in our proposed algorithm. Offline RL is fundamentally challenged by extrapolation errors and value overestimation, both of which stem from the tendency of value functions to over-generalize to out-of-distribution (OOD) actions. While in-sample learning approaches, such as Implicit Q-Learning (IQL) (Kostrikov et al., 2021), circumvent this issue by entirely restricting policy updates to the support of the empirical behavior policy, they forgo the potential benefits of controlled generalization beyond the observed data. This dichotomy gives rise to two opposing failure modes:

1. **Over-Generalization:** When value estimates are queried at OOD actions, erroneous predictions are propagated through Bellman backups, often leading to unstable or divergent learning dynamics.

2. **Under-Generalization (Non-Generalization):** Strict in-sample methods constrain the learned policy to lie within the convex hull of observed actions, thereby limiting policy improvement and potentially yielding suboptimal performance.

To reconcile these extremes, the Doubly Mild Generalization (DMG) framework was recently proposed by (Mao et al., 2024). DMG introduces a principled compromise that permits mild generalization while preserving robustness, by jointly regulating both action selection and value propagation.

The framework comprises two core mechanisms:

- **Mild Action Generalization:** The learned policy $\pi(\cdot\mid s)$ is allowed to explore a local neighborhood around the support of the empirical behavior policy $\mu(\cdot\mid s)$, formalized by the constraints:

$$\mathrm{supp}(\mu(\cdot\mid s)) \subseteq \mathrm{supp}(\pi(\cdot\mid s)), \tag{12}$$

$$\max_{a_1\sim\pi(\cdot|s)}\min_{a_2\sim\mu(\cdot|s)}\|a_1 - a_2\| \le \epsilon_a, \tag{13}$$

where $\epsilon_a > 0$ is a small tolerance parameter, and $\mu$ denotes the empirical behavior policy derived from the offline dataset.

- **Mild Generalization Propagation:** The Bellman update blends value estimates from both generalized and in-sample actions via a convex combination:

$$\mathcal{T}_{\text{DMG}} Q(s,a) := R(s,a) + \gamma \mathbb{E}_{s' \sim P(\cdot|s,a)} \left[ \lambda \max_{a' \sim \pi(\cdot|s')} [Q(s',a')] + (1-\lambda) \max_{a' \sim \mu(\cdot|s')} Q(s',a') \right], \quad (14)$$

where $\lambda \in [0,1]$ controls the trade-off between policy-driven generalization and conservative in-sample estimation. Notably, the first term enables mild optimism through policy-guided exploration, while the second term ensures robustness by anchoring the update to the empirical data distribution.

Under this design, DMG enjoys provable performance guarantees under both oracle and adversarial generalization regimes, as established in (Mao et al., 2024).

## E Conservative Estimation Analysis

This section provides a more detailed discussion of the conservative estimation introduced in 4.3.

To initiate the analysis, consider the quantity $Q(s',a')$ employed in Doubly Mild Generalization (DMG), where $s' \sim \mathcal{D}$ and $a' \sim \pi(\cdot \mid s')$. Since $(s',a')$ resides outside the support of the dataset, it is inherently excluded from the fitted Q-iteration (FQI) process. Accordingly, Assumption 1 is not imposed on $Q(s',a')$. Instead, this quantity is interpreted as the value $Q_\pi(s',a')$ under policy $\pi$. Therefore, it then follows that:

**Proposition 5.** *For a pair $(s',a')$ with $s' \sim \mathcal{D}$ and $a' \sim \pi(\cdot \mid s')$, there exists:*

$$Q(s',a') = Q_\pi(s',a') \leq Q_{\pi^\star}(s',a') = Q^\star(s',a').$$

In particular, $Q(s',a')$ serves as a conservative estimate of the optimal action-value function $Q^\star(s',a')$, i.e., $Q(s',a') \leq Q^\star(s',a')$ for all $(s',a')$. Moreover, when the policy $\pi$ is an improved, near-optimal policy, the degree of conservatism is expected to be relatively mild.

Recall that the optimal value function satisfies the identity

$$V^\star(s') = \beta(s') \log \int \exp\left( \frac{Q^\star(s',a')}{\beta(s')} \right) da'.$$

Since the exponential and logarithm are monotonic functions, replacing $Q^\star$ with its conservative approximation $Q$ yields a lower bound. Specifically, $\bar{V}(s') = \beta(s') \log \int \exp\left( \frac{Q(s',a')}{\beta(s')} \right) da'$ provides a conservative estimate of $V^\star(s')$, and by extension, of the empirical value estimate $\hat{V}(s')$. Under Assumption 2, $\bar{V}(s')$ can be approximated by the $\alpha_1$-quantile of $Q(s',a')$, yielding a mild yet principled conservative estimate of $\hat{V}(s')$.

In light of this reasoning, during the DMG, the target for $\hat{V}(s')$ is set to the $\alpha_1$-quantile of $Q(s',a')$. Accordingly, the $\alpha_0$-quantile of $Q(s',a')$ is employed as the target for $\hat{V}(s')$.

## F Baseline Hyperparameter Settings

### F.1 Extreme $Q$-Learning

This section illustrates the temperature hyperparameter $\beta$ settings used in the Extreme $Q$-Learning (XQL) baseline (Garg et al., 2023) across various offline reinforcement learning (RL) tasks from the D4RL and NeoRL2 benchmark. Dataset-specific $\beta$ values are listed in Tables 5, 6, 7 and 8, corresponding to the results in Table 1 and the first row of results in Table 3 in the experiment section. Domain-consistent $\beta$ settings are shown in Table 9 and relate to the second row of Table 3, while a fully consistent setting across

| Task (Variant) | halfcheetah | hopper | walker2d |
|---|---|---|---|
| medium | 1.0 | 5.0 | 10.0 |
| medium-rep | 1.0 | 2.0 | 5.0 |
| medium-exp | 1.0 | 2.0 | 2.0 |

Table 5: XQL temperature settings ($\beta$) with dataset-specific tuning on D4RL locomotion datasets.

| Task (Variant) | pen | hammer | door |
|---|---|---|---|
| human | 5.0 | 0.5 | 1.0 |
| cloned | 0.8 | 5.0 | 5.0 |

Table 6: XQL temperature settings ($\beta$) with dataset-specific tuning on D4RL Adroit datasets.

| Hyperparameter | antmaze-umaze | antmaze-umaze-diverse |
|---|---|---|
| $\beta$ | 1.0 | 5.0 |

Table 7: XQL temperature settings ($\beta$) with dataset-specific tuning on D4RL AntMaze datasets.

| Hyperparameter | RocketRecovery | SafetyHalfCheetah |
|---|---|---|
| $\beta$ | 2.0 | 2.0 |

Table 8: XQL temperature settings ($\beta$) with dataset-specific tunning on NeoRL2 datasets.

all domains uses $\beta = 2.0$, as reported in the third row. The variability in optimal $\beta$ values across tasks highlights XQL's sensitivity to its temperature parameter and the importance of tuning it carefully for each environment, as also emphasized in (Garg et al., 2023). For all other hyperparameter settings, we follow the XQL reproduction provided by (Gao & Rui, 2023).

| Hyperparameter | D4RL Locomotion | D4RL Adroit | D4RL AntMaze |
|---|:---:|:---:|:---:|
| $\beta$ | 2.0 | 5.0 | 0.6 |

Table 9: XQL consistent temperature settings ($\beta$) per domain.

### F.2 MXQL and Other Baselines

For MXQL (Omura et al., 2024), we adopt the original JAX implementation and use the exact hyperparameter settings as reported in the paper. For the remaining baseline algorithms, we use the reproduced results provided by CORL (Tarasov et al., 2022).

## G Implementation Details

This section provides a comprehensive overview of the hyperparameters and implementation specifics of our proposed QQL algorithm.

Our implementation of QQL is built upon the publicly available PyTorch implementation of Implicit Q-Learning (IQL) from CORL (Tarasov et al., 2022). We adopt the identical neural network architecture and retain the exact hyperparameter settings used for network updates in the original IQL implementation.

A crucial aspect of our approach is ensuring that $\beta(s)$ remains positive during policy optimization. To achieve this, we define $\beta(s)$ as the absolute value of $(\hat{V}_{\psi_2}(s) - V_{\psi_1}(s))/\omega$. For all other procedural steps, the direct value of $(\hat{V}_{\psi_2}(s) - V_{\psi_1}(s))/\omega$ is utilized. To prevent $\beta(s)$ from becoming infinitesimally small, we apply a lower bound clipping, setting $\beta(s) \geq \beta_{low} = 0.1$ specifically within the policy optimization routine. For consistency across all evaluated task-dataset combinations, we fix the generalization coefficient $\lambda$ to 1.0 and the policy constraint weight $\zeta$ to 1.0.

Table 10 summarizes the detailed hyperparameters employed in our QQL algorithm.

Table 10: Detailed Hyperparameters of the QQL Algorithm.

| Parameter | Value |
|---|---|
| V-function learning rate ($\alpha_V$) | $3 \times 10^{-4}$ |
| Q-function learning rate ($\alpha_Q$) | $3 \times 10^{-4}$ |
| Policy learning rate ($\alpha_\pi$) | $3 \times 10^{-4}$ |
| Discount factor ($\gamma$) | 0.99 |
| Target network update rate ($\tau$) | 0.005 |
| Batch size | 256 |
| $\beta_{low}$ | 0.1 |
| $\lambda$ | 1.0 |
| $\zeta$ | 1.0 |
| Hidden layer dimension | 256 |
| Number of hidden layers | 2 |
| Activation function | ReLU |

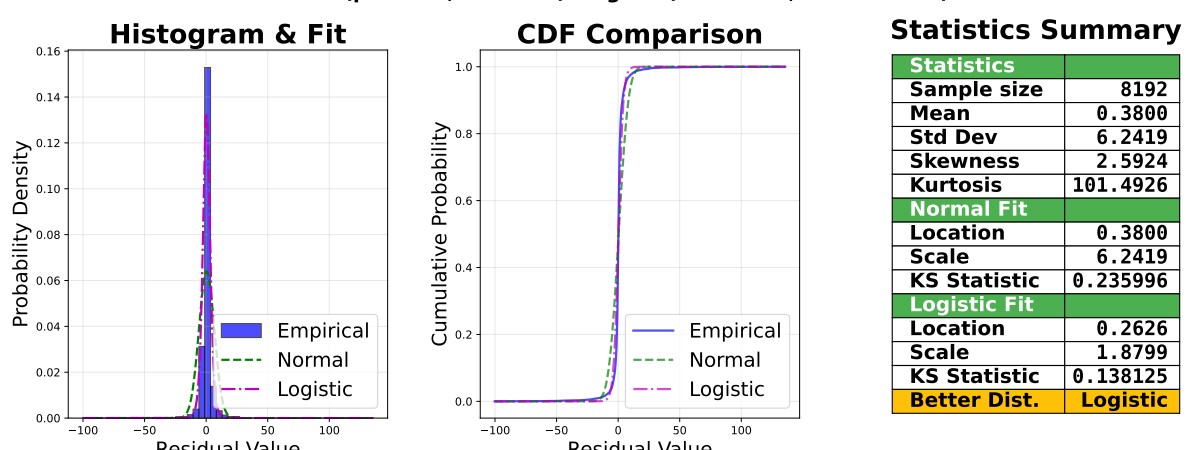

Figure 5: **Residual Analysis: HalfCheetah-Medium.**

Figure 6: **Residual Analysis: HalfCheetah-Medium-Replay.**

# H    Residual Analysis on D4RL

To empirically evaluate the plausibility of the modeling assumptions, a residual analysis is performed using value functions trained on the D4RL benchmark. For transition tuples $(s, a, r, s')$ sampled from the offline dataset, the regression residual is defined as

$$\epsilon(s, a, r, s') = Q_\theta(s, a) + \left(\hat{V}_{\psi_2}(s) - V_{\psi_1}(s)\right) - r(s, a) - \gamma\hat{V}_{\psi_2}(s'),$$

which measures the discrepancy between the predicted and target values under the assumed Bellman relationship. To mitigate the effect of state-dependent variance, residuals are normalized by a learned scale parameter $\beta(s)$, resulting in the normalized residual

$$\epsilon_{\text{norm}}(s, a, r, s') = \frac{\epsilon(s, a, r, s')}{\beta(s)}. \tag{15}$$

This normalized residual is subsequently analyzed across the D4RL locomotion tasks to assess the distributional behavior and potential systematic deviations of the estimated value functions.

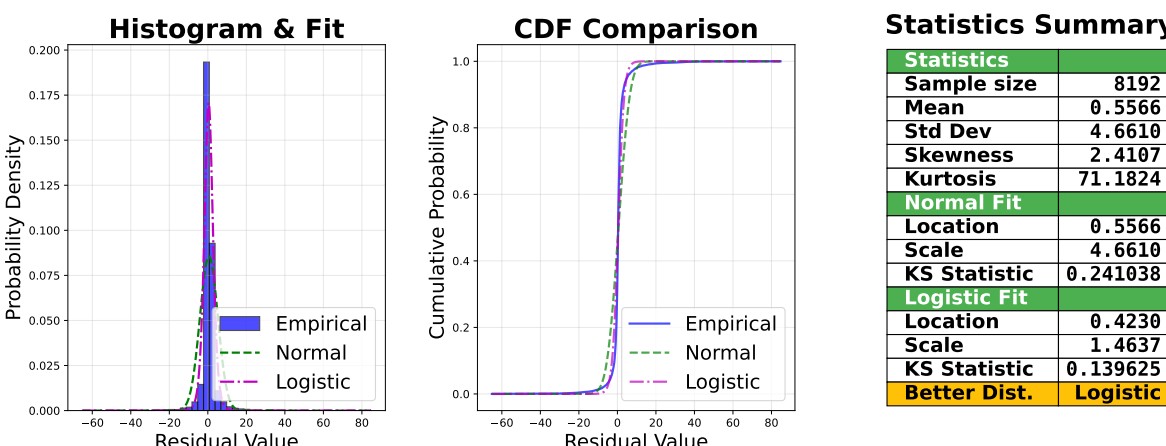

Figure 7: **Residual Analysis: Walker2d-Medium.**

Figure 8: **Residual Analysis: Walker2d-Medium-Replay.**

**Results and Discussion.** For the normalized residuals $\epsilon_{\text{norm}}$ on four D4RL tasks (Figures 5–8), we report maximum-likelihood fits under both normal and logistic distributional assumptions. The empirical histograms and cumulative distribution functions exhibit heavier tails and slight asymmetry compared to the normal fit, while the logistic fit aligns more closely with the observed data in these regions. Consistently across tasks, the logistic model yields lower Kolmogorov–Smirnov (KS) statistics than the normal model (e.g., 0.138 vs. 0.236 in HalfCheetah-Medium-Replay). These empirical findings suggest that, at least in comparison to the homoscedastic Gaussian assumption, the logistic noise model, as implied by Assumption 1 and derived in (Hui et al., 2023), provides a more realistic characterization of Bellman residuals in offline Q-learning.

