# OpenReview forum: "Quantile $Q$-Learning: Revisiting Offline Extreme $Q$-Learning with Quantile Regression"
_TMLR — Accepted by TMLR_

### Review · Reviewer_nvmQ · 2025-12-16

**Summary Of Contributions:**

This paper proposes Quantile Q-Learning (QQL), an improvement over Extreme Q-Learning (XQL) for offline RL. The key contribution is a quantile-regression based estimation of the temperature parameter $\beta$, which removes the need for dataset-specific tuning. Extensive experiments on D4RL and NeoRL2 show that QQL achieves more stable training and competitive or superior performance compared to XQL and MXQL, using a single consistent set of hyperparameters across domains

**Audience:**

Yes

**Audience Explanation:**

Researchers working on offline RL or hyperparameter design in RL would find this paper particularly relevant, especially those studying or building upon XQL.

**Claims And Evidence:**

Yes

**Claims Explanation:**

The claims are well supported by rigorous theoretical analysis and extensive experimental results, including comprehensive ablation studies.

**Requested Changes:**

1. The related work section is relatively brief. It would be beneficial to include a broader discussion of relevant literature, particularly distributional RL methods that also employ quantile regression. In addition, many offline RL methods discussed appear to predate 2023. The authors are encouraged to include more recent and representative offline RL methods, as well as a more comprehensive review of the XQL line of work.

2. Although the core idea of the paper is conceptually simple and many of the theoretical results are relatively straightforward, the overall presentation is difficult to follow, especially for readers who are not already familiar with XQL. Several aspects could be improved to enhance readability and accessibility:

- The authors should provide more detailed background on XQL, ideally in a dedicated subsection.

- In Section 3, the equations should be discussed in greater depth, including their intuitive meanings and practical implications. In particular, the paper should clearly explain the role of the temperature parameter $\beta$, why it is important, and why performance is sensitive to its choice.

- The intuitions behind Assumption 1 and Assumption 2 should be explained more clearly. Rather than only citing prior work, the authors should either summarize the underlying ideas from those papers or provide their own interpretation to ensure conceptual consistency and clarity.

3. In the main experimental results, it is unclear how the $\beta$ of XQL were tuned. The authors should clarify the tuning procedure, and explain how they ensured that the chosen $\beta$ were appropriate and fair for comparison.

---

> ### Author Response · Authors · 2026-01-09
>
> We thank Reviewer nvmQ for the thoughtful comments and for noting that our claims are "well supported by rigorous theoretical analysis and extensive experimental results". We are encouraged that the reviewer found our findings "particularly relevant" to the offline RL community. Below, we address each major concern in detail.
>
>
> **Related Work and Baseline Selection Logic (Addressing Requested Changes 1)**
>
> We have expanded the Related Work section to include a broader discussion of distributional RL and quantile regression, providing a more comprehensive review of the XQL lineage.
>
> Regarding the experimental baselines, We have already included a comprehensive set of model-free baselines. While we acknowledge the rapid emergence of generative (e.g., Diffusion-based) and model-based offline RL, we maintain that the primary contribution of QQL is a hyperparameter-robust improvement of XQL, it's unfair to compare QQL with these SOTA algorithms.
>
>
>
> **Enhancing Readability and Intuition (Addressing Requested Changes 2)**
>
> To improve the accessibility of our work for a broader audience and address the reviewer's concerns regarding the presentation of foundational concepts, we have made the following revisions:
>
> 1. **Detailed Background on XQL:**
> We have added a new appendix section **Appendix C: Analysis of the Error Model**, which presents a concise and self-contained overview of the theoretical foundations of Extreme Q-Learning (XQL). The section follows the derivation in [1] and classical results from [2]. It covers the Rust–McFadden latent-variable MDP formulation and the key assumptions of Conditional Independence and Additive Separability. It also introduces the Gumbel Error Model (GEM), which characterizes the evolution of the Q-value distribution under Gumbel-distributed Bellman errors. We hope this addition is helpful to readers who are less familiar with the theoretical underpinnings of XQL.
>
> 2. **Expanded Discussion of Key Equations and Intuitions in Section 3:**
> We have revised Section 3 to include a dedicated paragraph that explains the relationship between the temperature parameter $\beta$ and conservatism from the perspective of AWR[3], and to analyze the resulting parameter sensitivity issues in XQL. Specifically, the new text revisits the AWR objective to clarify how $\beta$ trades off exploiting high-valued actions and staying close to the behavior policy, and then connects this mechanism to the appearance of $\beta$ in the exponential terms of XQL’s value and policy updates, highlighting its impact on stability and hyperparameter sensitivity.
>
>
>
> 3. **Clarified Motivation and Interpretation of Assumption 1 and Assumption 2:**
> We have revised Section 4 to include **Remark on Assumption 1** to briefly explain the intuition behind Assumption 1, which is a more realistic assumption compared to homoscedastic Gaussian noise assumption. The intuition underlying Assumption 2 is now clarified in the **Remark on Assumption 2** and further elaborated in **Appendix C: Analysis of the Error Model**.
>
>
>
>
> **Clarification of Baseline Tuning and Fair Comparison (Addressing Requested Changes 3)**
>
>
> As discussed in **Appendix F: Baseline Hyperparameter Settings** of the revised manuscript, which corresponds to **Appendix C** in the original submission, we provide here a detailed description regarding our selection of baseline hyperparameters, particularly for the XQL method.
>
> We clarify that our $\beta$ choices for the XQL baseline strictly follow the protocol of the original work [1]. The original authors manually searched over candidate values and reported both **task-wise optimal** $\beta$ (per environment) and **domain-wise consistent** $\beta$ (per benchmark suite), which we directly adopt to reflect XQL’s intended performance.
>
> In our **main results**, we use the task-wise optimal $\beta$ reported by the original paper (“XQL Tuned”), so that XQL is evaluated under its best-known settings and serves as a strong, competitive baseline. In **Section 5.3**, we instead fix a single consistent $\beta$ across tasks to study sensitivity to this hyperparameter and to mimic more realistic settings where extensive per-task tuning is impractical, thereby highlighting the robustness of our method under limited tuning.
>
> [1] Divyansh Garg et al., 2023, Extreme Q-Learning: Maxent RL Without Entropy, arXiv:2302.05994.
>
> [2] Daniel McFadden,1972, Conditional logit analysis of qualitative choice behavior.
>
> [3] Xue Bin Peng et al.,2019, Advantage-weighted regression: Simple and
> scalable off-policy reinforcement learning,arxiv:1910.00177.

---

### Review · Reviewer_v2AK · 2025-12-27

**Summary Of Contributions:**

**Summary Of Contributions:** The paper addresses the instability and hyperparameter sensitivity of Extreme Q-Learning (XQL) in offline reinforcement learning (RL). It proposes a new algorithm, Quantile Q-Learning (QQL), which eliminates the need for dataset-specific tuning of the temperature parameter ($\beta$) in XQL. The authors derive a theoretical relationship allowing $\beta$ to be estimated as a state-dependent function using the difference between the expected state value and the SoftArgMax value. This estimation is implemented via quantile regression, where specific quantiles correspond to these value formulations under a Gumbel distribution assumption. Experimental results on D4RL and NeoRL2 benchmarks show that QQL with a single, consistent set of hyperparameters matches or outperforms XQL and other state-of-the-art baselines. To further improve training stability and prevent overestimation, the paper introduces Value Regulation (VR) and Conservative Estimation (CE), and demonstrate that the effectiveness of these mechanisms in the ablation studies.

**Strengths:**
- The most significant contribution is the elimination of the temperature hyperparameter $\beta$ tuning. Standard XQL is highly sensitive to $\beta$, often requiring a sweep of values (e.g., 1.0 to 10.0) for different datasets. QQL can use a fixed configuration across diverse domains (Locomotion, Adroit, AntMaze), which is a strong practical advantage for real-world deployment.
- The derivation connecting the temperature $\beta$ to the gap between specific quantiles of the Q-function (leveraging the Euler-Mascheroni constant) is sound. It provides a principled way to adapt the temperature locally per state rather than using a global heuristic.
- The experimental results on widely accepted benchmarks such as D4RL and NeoRL2 are strong and comprehensive, showing that QQL with fixed hyperparameters outperforms XQL even when XQL is tuned. Figure 2 also demonstrates its superior training stability compared to MXQL.
- The ablation studies in Figure 3 effectively isolate the benefits of the proposed Value Regulation and Conservative Estimation components. Figure 4 also illustrates the empirical distribution of the Q-function mirroring the Gumbel distribution, partially validating the assumption.

**Weaknesses:**
- The core derivation relies heavily on Assumptions 1 and 2, specifically that the error in Q-functions follows a Gumbel distribution and that the scale $\beta(s)$ is consistent across different formulations. While the authors provide a "Toy Example" to validate this, the complex dynamics of high-dimensional offline RL tasks may not preserve these distributional properties as cleanly as the toy example suggests.
- In Section 4.3, the explanation of Value Regulation with Mild Generalization is somewhat dense and relies on the reader being familiar with Doubly Mild Generalization (Mao et al., 2024). The rationale for why $\lambda=1$ is universally effective is empirically stated but not theoretically justified in depth within this text.
- The method requires learning two separate value networks ($V$ and $\hat{V}$) using quantile regression, plus the Q-network. An experiment comparing the computational cost of QQL to that of XQL or MXQL would be beneficial.
- (Minor) In Table 1, the QQL results are averaged over 5 random seeds with standard deviations, but the standard deviations of the baselines are missing.

**Additional Comments:**

N/A

**Audience:**

Yes

**Audience Explanation:**

Offline RL is a major subfield within machine learning. Methods that reduce hyperparameter sensitivity are highly valuable for practitioners who cannot afford to tune parameters in offline settings. The theoretical link between Extreme Value Theory and Quantile Regression is also of interest to the theoretical RL community.

**Claims And Evidence:**

Yes

**Claims Explanation:**

The paper demonstrates strong empirical evidence for its primary claim: that QQL achieves high performance without extensive hyperparameter tuning. The theoretical derivations in the main text and appendix appear sound based on the stated assumptions.

**Requested Changes:**

1. Deepen the discussion on Distributional Assumptions: Expand Section 5.5 or the limitations section to discuss what happens when the Gumbel assumption fails. Does the method degrade gracefully? If possible, provide a metric or visualization for a complex task (like AntMaze) similar to the Toy Example (Figure 4) to check if the Gumbel fit actually holds in deep feature spaces, not just in the toy example.
2. Clarify Value Regulation (Section 4.3):
- Provide a more self-contained explanation of "Mild Generalization" so readers do not need to read Mao et al. (2024) to understand the core idea.
- Explain intuition for why the specific target quantiles ($\alpha_0, \alpha_1$) were chosen for the regularization term beyond just "conservative estimation"
3. Computational Cost Analysis:
- Add an experiment, or theoretical justification compariing of training time/computational resources required for QQL versus standard XQL and MXQP.
4. Main experimental results in Table 1:
- Provide the standard deviations of the baselines results averaged over 5 random seeds.
5. Typos:
- Fix the reference to "Equation equation 9" in Appendix A.
- Correct "Ommited" to "Omitted" in Appendix B.

---

> ### Author Response · Authors · 2026-01-09
>
> We thank Reviewer v2AK for the thoughtful and constructive feedback on our work. We appreciate the positive assessment of our goal of reducing the instability and hyperparameter sensitivity of XQL in offline RL and respond to each concern in detail below.
>
>
> **Expanded Discussion of Distributional Assumptions and Empirical Validation (Addressing Requested Changes 1)**
> We performed a residual analysis on four D4RL MuJoCo datasets by normalizing the Q-update regression errors using the learned state-dependent scale $\beta(s)$ and pooling the normalized residuals. The empirical distribution of these residuals aligns more closely with a logistic distribution than with a homoscedastic Gaussian, as evidenced by consistently lower Kolmogorov–Smirnov (KS) statistics under the logistic fit. This observation is consistent with the theoretical noise model derived under the Gumbel error assumption in [1]. A new appendix section **Appendix H: Residual Analysis on D4RL** provides a detailed discussion of the analysis and results. To ensure readers are aware of this empirical validation, we have added a brief reminder in Section 5.5.
>
>
> **Clarified Value Regulation and “Mild Generalization” Intuition (Addressing Requested Changes 2)**
>
> 1. **Provide a more self-contained explanation of "Mild Generalization"**
> We added a new appendix section, **Appendix D: Doubly Mild Generalization**, which provides a concise, self-contained overview of DMG, including its motivation, the two components (mild action generalization and mild generalization propagation), and their theoretical guarantees — so readers can understand the core idea without consulting [2].
>
> 2. **Explain intuition for why specific target quantiles $\alpha_0$, $\alpha_1$ were chosen**
> A new appendix section, **Appendix E: Conservative Estimation Analysis**, has been added. It derives the principle for selecting the quantiles $\alpha_0$ and $\alpha_1$ from a pessimistic perspective aimed at mitigating extrapolation error on out-of-distribution state–action pairs.
>
> **Computational Cost Analysis and Empirical Comparison (Addressing Requested Changes 3)**
> We incorporated an additional experiment that directly compares the wall-clock training time of QQL and baseline methods under the same hardware and implementation settings. On a single RTX 4070, QQL requires approximately 4 hours for 1M training timesteps, whereas XQL and MXQL take about 2.5 hours for the same budget; we believe this overhead is reasonable in view of the improved stability and performance reported in our experiments. To ensure readers are aware of this trade-off, **we have added a brief discussion of the computational cost in Section 5.2 Main Results** of the revised manuscript.
>
> **Reported Standard Deviations in the Main Results Table (Addressing Requested Changes 4)**
> We have added the standard deviations (over 5 random seeds) for the two main baselines, XQL and MXQL in the main results table for completeness.
>
> | Domain            | Dataset                | XQL   | MXQL  | QQL (Ours)        |
> |-------------------|------------------------|-------|-------|-------------------|
> | Gym Locomotion    | hopper-med-exp         | 111.2 ± 0.9 | 110.7 ± 1.2| **112.5 ± 1.3** |
> |                   | hopper-med             | 74.2 ± 4.1  | **80.9 ± 1.4** | 77.3 ± 3.8        |
> |                   | hopper-med-rep         | 100.7 ± 1.7 | **102.7 ± 0.7** | 101.1 ± 2.1 |
> |                   | halfcheetah-med-exp    | 94.2 ± 0.9  | 92.1  ± 0.4| **94.3 ± 1.8**    |
> |                   | halfcheetah-med        | 48.3 ± 0.3  | 47.7  ± 0.1| **49.5 ± 0.3**    |
> |                   | halfcheetah-med-rep    | 45.2 ± 0.1  | 45.7  ± 0.2| **46.6 ± 0.3**    |
> |                   | walker2d-med-exp       | 112.7 ± 0.3 | 111.2 ± 0.2| **113.2 ± 0.3**   |
> |                   | walker2d-med           | 84.2 ± 0.5  | 83.8  ± 0.4| **85.2 ± 1.3**    |
> |                   | walker2d-med-rep       | 82.2 ± 4.9  | 83.6  ± 5.2| **90.2 ± 2.1**    |
> | Adroit            | pen-human              | 105.3 ± 5.5 | 122.1 ± 2.5| **128.3 ± 4.1**   |
> |                   | pen-cloned             | 112.6 ± 5.3 | **117.4 ± 16.1** | 115.2 ± 4.7 |
> |                   | door-human             | 13.2 ± 3.3  | 18.3 ± 5.1 | **21.1 ± 5.6**   |
> |                   | door-cloned            | 1.1  ± 0.5  | 1.8 ± 2.0  | **8.8 ± 4.1**    |
> |                   | hammer-human           | 7.3 ± 2.3   | **14.7 ± 3.5** | 8.4 ± 3.1    |
> |                   | hammer-cloned          | 1.1 ± 0.3  | 11.1 ± 2.5  | **8.0 ± 4.4**   |
> | AntMaze           | antmaze-umaze          | 90.3 ± 2.1  | 88.3 ± 2.1| 88.5 ± 6.1        |
> |                   | antmaze-umaze-diverse  | 77.2 ± 6.6  | 53.2 ± 9.7| **81.3 ± 4.1**    |

---

> ### Author Response · Authors · 2026-01-09
>
> **Corrected Typos and References (Addressing Requested Changes 5)**
> We thank the reviewer for carefully reading the manuscript and pointing out the typos. The erroneous reference to “Equation equation 9” in Appendix A and the misspelling “Ommitied” in Appendix B have both been corrected in the revised version.
>
> [1] David Yu-Tung Hui et al., 2023, Double Gumbel Q-Learning, NeurIPS 36, 2580–2616.
>
> [2] Yixiu Maoet al., 2024, Doubly mild generalization for offline reinforcement learning, NeurIPS 37, 51436–51473.

---

### Review · Reviewer_eKES · 2026-01-10

**Summary Of Contributions:**

The paper contributes a state-dependent temperature estimation method for XQL using quantile regression and integrates a value regularization technique. While it improves the "usability" of XQL by using consistent hyperparameters, the technical contribution is a combination of existing ideas (XQL + IQL-style quantiles + DMG).

**Audience:**

Yes

**Audience Explanation:**

researchers working on the practical deployment of offline RL and those interested in Extreme Value Theory in RL.

**Claims And Evidence:**

Yes

**Claims Explanation:**

The experimental results on D4RL and NeoRL2 support the claim of improved stability and hyperparameter robustness.

**Requested Changes:**

1.  Provide a comparison against "XQL + DMG" to isolate the contribution of the automatic $\beta(s)$ estimation.

2. Include a sensitivity analysis of the $\alpha_0, \alpha_1, \alpha_2$ quantile values.

3. Elaborate on the "1.6x wall-clock time". Provide a table comparing parameters and FLOPs across XQL, MXQL, and QQL.

---

> ### Author Response · Authors · 2026-01-19
>
> We thank Reviewer eKES for recognizing the contributions on state-dependent temperature estimation, value regularization, and improved usability of XQL. We appreciate the constructive suggestions and address the requested changes in details below.
>
> **Ablation Study on “XQL + DMG” (Addressing Requested Changes 1)**
>
> To better isolate the contribution of the automatic $\beta(s)$ estimation from that of DMG, we have added a new set of ablation studies comparing QQL to **XQL-DMG** under the same training and evaluation protocol, as requested by the reviewer. We denote **XQL-DMG-T** and **XQL-DMG-C** as XQL-DMG with dataset-specific tuned and domain-specific tuned $\beta$, respectively. The specific $\beta$ values used for these baselines are detailed in the table below, following the hyperparameter protocol from Appendix F: Baseline Hyperparameter Settings in our revised manuscript (originally Appendix C). Across the representative D4RL Gym Locomotion tasks shown in the performance table, the proposed adaptive $\beta(s)$ estimation in QQL (with consistent hyperparameters across all datasets and domains) achieves performance that is competitive with, and in several cases better than, the carefully tuned XQL-DMG variants, demonstrating the effectiveness of our proposed estimation approach with quantile regression.
>
> | Domain            | Dataset              | XQL-DMG-T      | XQL-DMG-C      | QQL (Ours)        |
> |-------------------|---------------------|----------------|----------------|-------------------|
> | Gym Locomotion    | halfcheetah-med     | 47.9 ± 0.1     | 47.7 ± 0.3     | 49.5 ± 0.3        |
> |                   | walker2d-med        | 84.6 ± 1.3     | 85.8 ± 1.0     | 85.2 ± 1.3        |
> |                   | hopper-med-rep      | 101.5 ± 0.4    | 101.5 ± 0.4    | 101.1 ± 2.1       |
> |                   | halfcheetah-med-rep | 45.3 ± 0.6     | 45.5 ± 0.3     | 46.6 ± 0.3        |
>
> **Hyperparameter Settings for XQL-DMG Baselines:**
>
> | Domain            | Dataset              | $\beta$ for XQL-DMG-T | $\beta$ for XQL-DMG-C |
> |-------------------|---------------------|-----------------------|-----------------------|
> | Gym Locomotion    | halfcheetah-med     | 1.0                   | 2.0                   |
> |                   | walker2d-med        | 10.0                  | 2.0                   |
> |                   | hopper-med-rep      | 2.0                   | 2.0                   |
> |                   | halfcheetah-med-rep | 1.0                   | 2.0                   |
>
>
>
>
> **Sensitivity Analysis of $\alpha_0, \alpha_1, \alpha_2$ Quantiles (Addressing Requested Changes 2)**
>
> In Section 4, we provide a detailed theoretical justification for the choice of quantiles $\alpha_0$, $\alpha_1$, and $\alpha_2$. These quantiles have clear theoretical interpretations and are not treated as tunable hyperparameters.
>
> We provide additional experiments to demonstrate that minor perturbations to these quantiles near the true value have small impact on QQL's performance. Specifically, recognizing the interdependence among $\alpha_0$, $\alpha_1$, and $\alpha_2$, we parameterize them as $\alpha_1 = 1 - \exp(-\exp(\mu))$, $\alpha_0 = 1 - \exp(-\exp(\mu - \omega))$, $\alpha_2 = 1 - \exp(-\exp(\mu + \omega))$, where $\mu$ serves as an adjustable location parameter and $\mu = 0$ recovers the standard QQL configuration. We tested $\mu = \{0.1, -0.1\}$ against the baseline ($\mu = 0$) on representative tasks, with results shown below:
>
> | Dataset              | QQL ($\mu$ = 0)  | QQL ($\mu$ = 0.1) | QQL ($\mu$ = -0.1) |
> |----------------------|------------------|-------------------|--------------------|
> | hopper-med-exp       | 112.5 ± 1.3       | 111.9 ± 1.3       | 112.3 ± 0.2         |
> | halfcheetah-med      | 49.5 ± 0.3      | 49.6 ± 0.1        | 49.2 ± 0.1         |
> | walker2d-med-rep     | 90.2 ± 2.1       | 90.6 ± 0.8        | 89.5 ± 0.6         |
>
> These empirical results confirm that slight quantile shifts have negligible effects on performance, demonstrating the robustness of our proposed approach.

---

> > ### Author Response · Authors · 2026-01-19
> >
> > **Computational Cost Analysis (Addressing Requested Changes 3)**
> >
> > We elaborate on the "1.6× wall-clock time" observation by providing detailed measurements across four representative D4RL tasks: `halfcheetah-medium-replay-v2`, `walker2d-medium-replay-v2`, `halfcheetah-medium-v2`, and `walker2d-medium-v2`. All experiments used identical settings (RTX 4070 GPU, batch_size=256, PyTorch 2.1.0+cu121) for 1M training timesteps. The results are summarized in Tables 1-4 below, comparing XQL, MXQL, and QQL across wall-clock time, trainable parameters, and FLOPs with all networks summed and actor only (during inference), respectively. All FLOPs reported in this section correspond to forward-pass computations only.
> >
> > **Table 1: Wall-Clock Training Time (for 1M timesteps)**
> >
> > | Task                    | XQL          | MXQL         | QQL          |
> > |-------------------------|--------------|--------------|--------------|
> > | halfcheetah-med-rep     | 2h 27m 19s   | 2h 31m 1s    | 4h 3m 7s |
> > | walker2d-med-rep        | 2h 28m 51s   | 2h 33m 47s   | 4h 9m 44s|
> > | halfcheetah-med         | 2h 27m 31s   | 2h 28m 54s   | 4h 8m 16s|
> > | walker2d-med            | 2h 26m 7s    | 2h 34m 26s   | 4h 4m 47s|
> >
> > **Table 2: Trainable Parameters (millions)**
> >
> > | Task                    | XQL   | MXQL  | QQL   |
> > |-------------------------|-------|-------|-------|
> > | halfcheetah-med-rep     | 0.29  | 0.29  | 0.36 |
> > | walker2d-med-rep        | 0.29  | 0.29  | 0.36 |
> > | halfcheetah-med         | 0.29  | 0.29  | 0.36 |
> > | walker2d-med            | 0.29  | 0.29  | 0.36 |
> >
> > **Table 3: FLOPs - All Networks Summed (millions)**
> >
> > | Task                    | XQL   | MXQL  | QQL   |
> > |-------------------------|-------|-------|-------|
> > | halfcheetah-med-rep     | 0.57  | 0.57  | 0.71 |
> > | walker2d-med-rep        | 0.57  | 0.57  | 0.71 |
> > | halfcheetah-med         | 0.57  | 0.57  | 0.71 |
> > | walker2d-med            | 0.57  | 0.57  | 0.71 |
> >
> > **Table 4: FLOPs - Actor Only (millions)**
> >
> > | Task                    | XQL   | MXQL  | QQL   |
> > |-------------------------|-------|-------|-------|
> > | halfcheetah-med-rep     | 0.14  | 0.14  | 0.14 |
> > | walker2d-med-rep        | 0.14  | 0.14  | 0.14 |
> > | halfcheetah-med         | 0.14  | 0.14  | 0.14 |
> > | walker2d-med            | 0.14  | 0.14  | 0.14 |
> >
> > **Result Interpretation**:
> > - **Training Time**: QQL exhibits ~1.6× wall-clock time compared to XQL across all tasks, primarily due to (1) the additional Doubly Mild Generalization (DMG) computation steps and (2) extra backward passes through the additional V-network.
> > - **Parameters**: QQL has ~24% more trainable parameters (0.36M vs 0.29M) due to the additional V-network.
> > - **FLOPs - All Networks Summed**: Total FLOPs increase by ~25% (0.71M vs 0.57M), reflecting the cumulative forward passes across all networks (2 Q networks, V network and Actor network).
> > - **Inference Efficiency**: Inference FLOPs remain identical across all methods (0.14M), as deployment only requires the actor network, which is the same across all baselines.
> >
> > These results confirm QQL's modest computational overhead compared to existing baselines.

---

### Author Response · Authors · 2026-01-09

Dear Reviewers,

We sincerely thank you for your time and for providing constructive and insightful feedback on our submission. In response to your comments and concerns, we have revised the manuscript to include additional clarifications and new experimental results where requested. All newly added or substantially revised content is highlighted in blue in the updated manuscript for ease of reference.

Thank you again for your valuable feedback and consideration.

Best regards,

The Authors

---

### Decision · Action_Editor_wUB6 · 2026-04-02

**Recommendation:** Accept with minor revision

**Audience:**

Yes

**Audience Explanation:**

Researchers from offline RL will be interested in this work.

**Claims And Evidence:**

Yes

**Claims Explanation:**

The paper claims to provide a principled approach to estimate the temperature coefficient in XQL and MXQL. Under some Gumbel assumptions, an estimation is provided, and following algorithm is designed and analyzed. The experiment results also validate the claim.